# Is the Arginase Pathway a Novel Therapeutic Avenue for Diabetic Retinopathy?

**DOI:** 10.3390/jcm9020425

**Published:** 2020-02-05

**Authors:** Esraa Shosha, Abdelrahman Y. Fouda, S. Priya Narayanan, R. William Caldwell, Ruth B. Caldwell

**Affiliations:** 1Vascular Biology Center, Augusta University, Augusta, GA 30912, USA; eshosha@augusta.edu (E.S.); afouda@augusta.edu (A.Y.F.); pnarayanan@augusta.edu (S.P.N.); 2Clinical Pharmacy Department, Faculty of Pharmacy, Cairo University, Cairo 11562, Egypt; 3Vision Discovery Institute, Augusta University, Augusta, GA 30912, USA; wcaldwel@augusta.edu; 4Charlie Norwood VA Medical Center, Augusta, GA 30912, USA; 5Program in Clinical and Experimental Therapeutics, College of Pharmacy, University of Georgia, Augusta, GA 30912, USA; 6Department of Pharmacology & Toxicology, Augusta University, Augusta, GA 30912, USA

**Keywords:** diabetic retinopathy, arginase, neurovascular injury, therapy

## Abstract

Diabetic retinopathy (DR) is the leading cause of blindness in working age Americans. Clinicians diagnose DR based on its characteristic vascular pathology, which is evident upon clinical exam. However, extensive research has shown that diabetes causes significant neurovascular dysfunction prior to the development of clinically apparent vascular damage. While laser photocoagulation and/or anti-vascular endothelial growth factor (VEGF) therapies are often effective for limiting the late-stage vascular pathology, we still do not have an effective treatment to limit the neurovascular dysfunction or promote repair during the early stages of DR. This review addresses the role of arginase as a mediator of retinal neurovascular injury and therapeutic target for early stage DR. Arginase is the ureohydrolase enzyme that catalyzes the production of L-ornithine and urea from L-arginine. Arginase upregulation has been associated with inflammation, oxidative stress, and peripheral vascular dysfunction in models of both types of diabetes. The arginase enzyme has been identified as a therapeutic target in cardiovascular disease and central nervous system disease including stroke and ischemic retinopathies. Here, we discuss and review the literature on arginase-induced retinal neurovascular dysfunction in models of DR. We also speculate on the therapeutic potential of arginase in DR and its related underlying mechanisms.

## 1. Introduction

According to the latest national report on diabetes statistics published by the Centers for Disease Control and Prevention, 9.4% of the US population is diabetic [1]. Diabetes puts a large burden on the US economy. The total estimated direct and indirect cost of diagnosed diabetes in 2012 was $245 billion [2]. Diabetic retinopathy is one of the most devastating microvascular complications of diabetes. It mainly affects working age adults and is the leading cause of blindness in that age group [3]. Recent population-based studies reported that 2.6 million people were visually impaired due to DR in 2015 and this number is expected to increase to 3.2 million people in 2020 [4,5].

Diabetic retinopathy (DR) is a neurovascular disease and is classified in two stages based on the vascular pathology; the early stage is non-proliferative diabetic retinopathy (NPDR) and the advanced stage is proliferative diabetic retinopathy (PDR) [6]. NPDR is usually asymptomatic, however microaneurysms are evident upon ophthalmoscopic examination and leakage of small vessels may cause the retina to swell, resulting in blurry vision. PDR is characterized by the formation of abnormal blood vessels on the surface of the retina. These new vessels are fragile and can leak fluid or blood into the vitreous. While DR is diagnosed based on the characteristic vascular pathology, neuronal injury is also a prominent feature and may precede the vascular damage [7,8,9]. Current therapies for DR include focal or pan retinal laser photocoagulation, vitrectomy surgery, and intravitreal injections of vascular endothelial growth factor (VEGF) inhibitors [10]. These treatments can be effective in limiting the late stage vascular pathology. However, these treatments are not effective for every patient and they are associated with risks of complications. In particular, anti-VEGF agents have been linked to adverse effects on the photoreceptors and choroidal vessels as well as on the kidney and cardiovascular system [11,12,13]. Moreover, none of these treatments addresses neuronal damage or promotes tissue repair. Thus, there is a great need for a better understanding of the molecular mechanisms underlying the development and progression of DR in order to identify new therapies to target the early aspects of the pathology.

There are a number of novel avenues being explored for treatment of DR by using strategies to stimulate the action of endogenous protective mechanisms [14]. These include enhancing the functions of superoxide dismutase 2 (MnSOD), pigment epithelium–derived factor (PEDF), somatostatin, brain derived neurotrophic factor (BNDF), nerve growth factor (NGF), and NF-E2–related factor 2 (Nrf2). These molecules can promote a variety of protective pathways in DR, including reducing oxidative stress (MnSOD and Nrf2), inflammation (PEDF and Nrf2), and neurodegeneration (somatostatin, BNDF, and NGF). Activation of peroxisome proliferator-activated receptor alpha (PPARα) can improve several aspects of DR, including reducing inflammation and vascular permeability. Cell-based strategies including endothelial progenitor cells and mesenchymal stem cells are also under consideration for their beneficial effects in promoting vascular repair and alleviating retinal ischemia. Recent studies have demonstrated the effective use of gene therapy to downregulate VEGF by targeting sFlt-1, Flt23k, and PEDF [15].

The critical role of oxidative stress and inflammation in DR has been well established by studies in a variety of experimental models and patient samples [9,16,17,18,19,20,21]. Clinical investigations have shown some promise of using inhibitors of oxidative stress to limit DR but so far, the treatments have been only partially effective and/or accompanied by adverse side effects [22,23,24,25]. One possible explanation for these disappointing results is a lack of specificity of the general antioxidants used in such trials. Thus, there is a critical need to identify specific up-stream pathways. Numerous studies in diabetic patients and a variety of experimental animal models have demonstrated the role of alterations in L-arginine metabolism mediated by upregulation of the urea cycle enzyme arginase in diabetes-induced oxidative stress, inflammation, and vascular dysfunction [26,27,28,29,30]. Arginase is a ureohydrolase that catalyzes the last step of the urea cycle in the liver to dispose of ammonia resulting from protein catabolism by converting L-arginine to urea and L-ornithine. It is constitutively active in most contexts, so an increase in its expression is usually accompanied by an increase in its activity. The sections that follow will develop the concept of alterations in L-arginine metabolism due to excessive expression/activity of arginase as a key mediator of oxidative stress, inflammation, and neurovascular dysfunction and injury during DR.

To our knowledge, L-arginine metabolism in human retinas has not been examined so far. However, metabolomics studies of vitreous humor samples collected from patients with PDR have shown significant increases in L-arginine metabolism as compared with samples from non-diabetic patients [31]. This alteration was accompanied by a prominent increase in L-proline, which is a downstream product of L-arginine metabolism by arginase. Further support for an association between diabetes-induced alterations in arginase-mediated L-arginine metabolism and DR is provided by metabolomics studies of plasma samples. Comparisons of plasma metabolic profiles from type 2 diabetic patients showed dysregulation of the L-arginine pathway in patients with PDR as compared with those with NPDR [32]. A similar metabolomics profiling study performed with plasma samples from type 2 diabetic patients in China found similar impairment in the metabolism of L-arginine and L-proline in patients with PDR as compared with diabetic patients without retinopathy [33].

## 2. Arginase

### 2.1. Arginase Isoforms and L-Arginine Metabolism

The arginase enzyme exists in two isoforms, arginase 1 (A1) and arginase 2 (A2) [34]. The two isoforms expressed by different genes which are localized on different chromosomes. Their amino acid sequences are about 60% homologous in humans but the catalytic site is identical in both isoforms. While both isoforms hydrolyze the semi-essential amino acid L-arginine to form L-ornithine and urea, they are localized to different intracellular compartments; A1 is cytosolic and is strongly expressed in the liver where it plays a key role in ammonia detoxification in the urea cycle. A2 localizes mainly to the mitochondria and is highly expressed in the kidney. However, both isoforms are widely expressed in vascular, immune, and neuronal cells [30,35,36]. Both A1 and A2 are expressed in the retina and are found in different layers and cell types [37]. A1 immuno-reactivity is prominent in neuronal cells within the ganglion cell layer, inner nuclear layer, and in Müller glial cells. Prominent A2 immunoreactivity is evident in cells of the inner nuclear and nerve fiber layers as well as in horizontal cells [37,38]. However, given the mitochondrial localization of A2, it is likely that it is present in all cell types.

Genetic studies have demonstrated a key role for both A1 and A2 in regulating L-arginine availability. Insufficiency of either isoform results in hyperargininemia [39,40,41]. However, the phenotype of A1-deficient mice is much more severe than that of A2-deficient mice. A1 null mice develop hyperammonemia and die shortly after birth due to the crucial role of A1 in detoxifying ammonia in the liver. In contrast, the A2 null mice do not have any phenotype except for a mild hypertension at 8–10 weeks of age [42]. These findings suggest that while both isoforms of arginase play an essential role in regulating circulating L-arginine levels, their biological actions are very different. Moreover, the lethal effects of the A1 deletion indicate that A2 expression does not compensate for a lack of A1, at least within the liver. Furthermore, genetic studies in a variety of disease models have shown opposite effects of A1 vs. A2 deletion, suggesting a lack of redundancy/compensation between the two isoforms. This evidence is outlined in Section 3 below.

L-arginine metabolism is a complex, yet well-studied process. As noted above, arginase metabolizes L-arginase to produce L-ornithine and urea (Figure 1). L-ornithine is the substrate for the ornithine decarboxylase (ODC) pathway for the production of different polyamines through sequential reactions (Figure 1) [43]. Polyamines are important for cell proliferation. L-ornithine is also a substrate for another enzyme called ornithine aminotransferase (OAT) and is utilized for the production of proline, which is a critical component for collagen formation. L-arginine is also a substrate used by nitric oxide synthase (NOS) for the production of nitric oxide and L-citrulline. L-citrulline can be recycled back to L-arginine by the consecutive enzymatic reactions of argininosuccinate synthase (ASS) and argininosuccinate lyase (ASL) (Figure 1) [43]. In the liver and the lining of the gastrointestinal tract, L-ornithine can be recycled back to L-arginine as well through the enzymatic reactions of ornithine transcarbamylase (OTC), ASS, and ASL [44].

### 2.2. L-Arginine Paradox

Even though L-arginine is a common substrate for both NOS and arginase, the two enzymes have different affinities for L-arginine. The affinity of NOS for L-arginine is much higher (Km = 2–20 µM) than the affinity of arginase (Km = 2–20 mM), but NOS and arginase use L-arginine at similar rates [45,46]. The reason behind this is that the Vmax of arginase is 1000 times higher than the Vmax of NOS [45,46].

NOS has three different isoforms: endothelial NOS (eNOS), neuronal NOS (nNOS), and inducible NOS (iNOS). The eNOS isoform is the best studied in relation to arginase. The concentration of L-arginine in the blood and within endothelial cells is 50 and 800 µM, respectively [47,48]. Despite the fact that eNOS is saturated with L-arginine and the intracellular concentration of L-arginine highly exceeds the amount needed for NO production, exogenous administration of L-arginine still can increase NO production in a phenomenon referred to as “the arginine paradox” [49]. One possible explanation for this phenomenon is that eNOS is co-localized with arginase and that arginase depletes the L-arginine supply locally, decreasing its availability to eNOS, thereby limiting NO production [50]. It has also been suggested that there are poorly interchangeable intracellular pools of L-arginine (Figure 2). If so, replenishment of the intracellular supply might require L-arginine recycling or its intracellular transport by the cationic amino acid transporter 1 (CAT-1). Therefore, the arginine paradox phenomenon could be partially explained by the local depletion of L-arginine [51]. Another explanation for the limited L-arginine bioavailability could be the impaired uptake and transport of L-arginine by CAT-1 (Figure 2). This may happen through reduced expression or function of CAT-1 or through increases in L-ornithine, which competes, with L-arginine for CAT-1 [52,53]. Nevertheless, studies have shown that the chronic administration of L-arginine does not increase NO production or improve endothelial function. Conversely, the chronic administration of L-arginine has been shown to increase the products of the arginase pathway: urea and L-ornithine, but not the products of the NOS pathway [54].

Given the well-known function of nNOS in retinal neuronal signaling and extensive evidence of iNOS activation in retinopathy, studies on the L-arginine paradox in relation to nNOS and iNOS are highly relevant to retinal diseases including DR. While little is known about nNOS in this context, a few studies have examined the relationship between L-arginine and iNOS within the central nervous system (CNS). The concentration of L-arginine in the pericellular space of the brain and cerebrospinal fluid is 20–40 μM [55]. Under inflammatory conditions, L-arginine transport into astrocytes is required for the maximal levels of iNOS activity [56]. Therefore, iNOS activity in astrocytes requires the extracellular availability of L-arginine, even though the intracellular concentration of this substrate is well above its Km. A study by Lee et al. has demonstrated that L-arginine not only controls iNOS activity by limiting its substrate availability, but also regulates iNOS mRNA translation [57]. Thus, increased levels of arginase expression could be a tool for limiting inflammation in the retina and brain by limiting the L-arginine supply for iNOS activity and mRNA translation.

**Figure 2 jcm-09-00425-f002:**
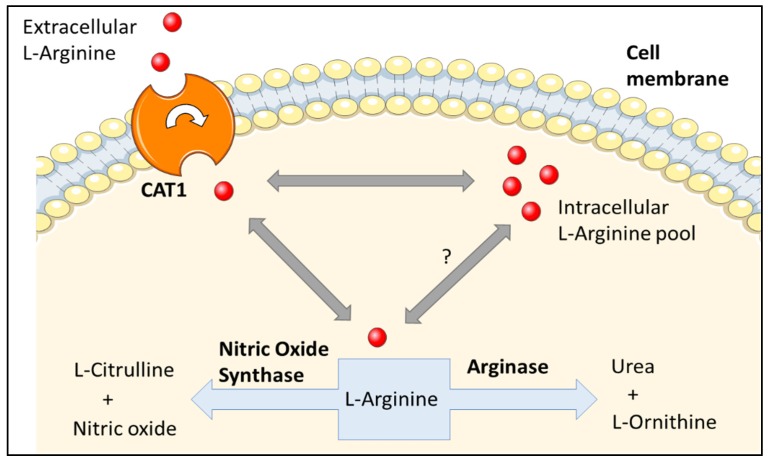
Schematic diagram of the L-arginine uptake and utilization by the cell. L-arginine is transported by the cationic amino acid transporter (CAT1). Inside the cell, L-arginine is thought to be present in poorly interchangeable pools that can be slowly depleted or replenished, thus accounting for the arginine paradox. The diagram was created using Servier® medical art https://smart.servier.com/ [58].

### 2.3. Arginase, NOS Uncoupling, Oxidative/Nitrative Stress, and Inflammation

As noted above, NO is produced by the three different isoforms of NOS: eNOS, nNOS, and iNOS. All three utilize L-arginine for the production of NO and L-citrulline (Figure 3A). This enzymatic reaction requires the co-factor tetrahydrobiopterin (BH4). The role of NO in maintaining vascular health is very well established. NO acts in an autocrine and paracrine fashion to regulate vascular tone, neurotransmission, immune responses, and oxidative stress [30]. NO produced by eNOS induces vasodilation and inhibits platelet aggregation and leukocyte adhesion to the vascular wall which limits vascular inflammation. In addition, eNOS-derived NO can affect smooth muscle cell proliferation and post-natal angiogenesis and activate endothelial progenitor cells [30,59]. On the other hand, nNOS-derived NO plays a key in the actions of nitrergic neurons on smooth muscle, gastrointestinal tract, and erectile tissue. In the CNS, NO can act as a neurotransmitter as well as to modulate neurotransmitter release. However, under inflammatory conditions when NO is produced in excess by iNOS and in the presence of superoxide anion, it can promote neurodegeneration due to the formation of the toxic oxidant peroxynitrite [60].

NOS can be uncoupled under conditions of limited availability of its substrate L-arginine or its co-factor BH4 [59]. As mentioned earlier, arginase competes with NOS for L-arginine, which limits NO production and causes NOS uncoupling [61,62]. When NOS is uncoupled, it uses molecular oxygen to produce superoxide anion (Figure 3B). Superoxide reacts rapidly with the available NO to form the toxic oxidant peroxynitrite. This further aggravates the oxidative stress condition. Thus, excessive expression of the arginase enzyme is a prominent source of oxidative stress. In addition to its actions in mediating oxidative stress, decreases in NO due to NOS uncoupling can contribute to inflammation via increases in platelet aggregation and leukocyte attachment to the vascular endothelium.

### 2.4. Arginase and Polyamine Metabolism

Arginase utilizes L-arginine for the production of L-ornithine, which is the precursor for production of polyamines and L-proline. Polyamines (spermine, spermidine, and putrescine) are important for cell proliferation, ion channel function, and neuroprotection. L-proline is required for collagen formation. At the same time, during the polyamine production cascade, multiple ROS are generated as by-products. Hydrogen peroxide, acrolein, and amino aldehydes are generated during polyamine oxidation via spermine oxidase (SMOX). SMOX is an inducible enzyme in the polyamine catabolism pathway. It catalyzes the oxidation of spermine to spermidine. However, at the same time, hydrogen peroxide and aldehyde 3-aminopropanal (3-AP) are generated as by-products. Both can induce cell damage through affecting DNA, RNA, proteins, and membranes [63]. 3-AP can be spontaneously converted to the toxic aldehyde acrolein, which causes protein modifications by reacting with nucleophilic sites [64]. Nε-(3-formyl-3,4-dehydropiperidino) lysine adduct (FDP-Lys), an acrolein-lysine adduct, is one such form of acrolein conjugation [65].

## 3. Arginase and Diabetic Retinopathy

### 3.1. Arginase in Diabetic Complications

The role of arginase in the systemic complications of diabetes is well established and has been extensively reviewed [27,29,30,66]. Studies in a model of type 1 diabetes have shown increased arginase expression along with impaired vasorelaxation of coronary arteries [61]. High glucose treatment of bovine aortic endothelial cells induced arginase activity along with increased ROS and decreased NO production. The inhibition of arginase activity blunted these diabetes or high glucose-induced dysfunctions. Arginase was also shown to contribute to diabetes-induced injury of cardiomyocytes by similar mechanisms involving increased NOS uncoupling and oxidative stress [67]. In a study looking at aortic rings from diabetic mice, the double knockout of A1 (one copy) and A2 (both copies) significantly protected against diabetes-induced impaired endothelial dependent vasorelaxation and vascular stiffness [68]. Excessive arginase has also been also implicated in other complications of both type 1 and type 2 diabetes, including nephropathy, neuropathy, and erectile dysfunction [29,69,70,71].

Upregulation of arginase expression/activity has also been strongly implicated in peripheral vascular and metabolic dysfunctions in a mouse model of Western diet-induced obesity and type 2 diabetes. EC-specific deletion of A1 or pharmacological inhibition of arginase activity blocked both obesity-induced increases in oxidative stress and impairment of endothelial-dependent relaxation but had no significant effect on obesity or metabolic dysfunction [72]. In contrast, studies using A2 null mice have shown significant protection against Western diet-induced obesity, visceral adipose tissue inflammation, and metabolic dysfunction as compared with wild type mice [73]. Interestingly, unlike the EC-specific A1 deletion, the A2 deletion was only modestly protective against the diabetes-induced vascular dysfunction. These differences highlight the differential roles of the two isoforms in the pathophysiology of diabetic complications.

Further evidence for the role of arginase in diabetic complications has been provided by studies showing that treatment of type 2 diabetic patients with the arginase inhibitor N-hydroxy-nor-L-arginine prevented the diabetes-induced impairment of endothelium-dependent vasorelaxation [28]. Also, studies of coronary arterioles from patients with or without diabetes showed that endothelium-dependent vasorelaxation was impaired in the vessels from the diabetic patients and that this was associated with increased endothelial cell expression of A1 [74].

Somewhat less is known about the involvement of arginase in diabetic retinopathy. However, a study using streptozotocin (STZ)-induced diabetic mice found significant increases in retinal arginase activity and A1 expression after 2 months of hyperglycemia or after high glucose (HG) treatment of retinal endothelial cells (RECs) [37]. The double knockout of one copy of A1 and both copies of A2 in diabetic mice or inhibition of arginase in RECs significantly reduced production of superoxide, confirming the role of arginase in diabetes- or HG-induced oxidative stress. Moreover, the beneficial effects of arginase inhibition in these diabetic models were accompanied by enhanced NO formation and decreased leukocyte adhesion, implying involvement of NOS uncoupling in the pathological process [37]. In addition, another study using both ex vivo and in vivo models found that diabetes impairs endothelial-dependent vasorelaxation of retinal vessels and that heterozygous deletion of A1 or arginase inhibition significantly protected against the diabetes-induced retinal vascular dysfunction [75]. More recently, a study in a mouse model of type 2 diabetes has shown the involvement of A2 in obesity-induced retinal injury. Mice fed Western diet for 16 weeks had significantly increased levels of A2 in their retinas. This was associated with abnormal or exaggerated photoreceptor light responses. The deletion of A2 significantly protected against this obesity-induced photoreceptor abnormality. This protective effect of the A2 deletion was accompanied by reduced retinal inflammation, oxidative stress, and microglia/macrophage activation [76]. Further study is required to elucidate the differential roles of the two arginase isoforms in the various retinal complications of type 1 and type 2 diabetes.

### 3.2. Arginase in Other Ischemic Retinopathies

Due to the fact that murine diabetic models do not develop vitreoretinal neovascularization or severe neuronal injury characteristic of late stage DR, many investigators have used other murine models of ischemic retinopathy to study retinal neovascularization and advanced stage neurodegeneration [77]. Oxygen-induced retinopathy (OIR) is one well-accepted model for severe retinal neurovascular injury [78]. In this model, neonatal mice are maintained in hyperoxia to cause degeneration of the developing vessels. This results in relative hypoxia when the mice are returned to a normoxic environment, which promotes vitreoretinal neovascularization. Studies have shown that levels of the mitochondrial A2 isoenzyme are increased during the ischemic phase of OIR along with increases in oxidative/nitrative stress, elevated iNOS expression, decreases in physiological vascular repair, and marked vitreoretinal neovascularization [79]. The vascular pathology was accompanied by increased neuronal cell death, glial activation, and impaired retinal function. Each of these alterations was significantly ameliorated by the global deletion of A2 [38,79,80]. Consistent with the apparent involvement of A2 in limiting vascular repair in the OIR model, studies using tissue culture cells showed that inhibition of angiogenesis during prolonged hypoxia is associated with increases in A2 expression and oxidative stress as well as decreases in NO formation and VEGF expression. Each of these effects was prevented by inhibition of arginase [81]. Given that eNOS activity in forming NO plays a key role in VEGF-induced increases in angiogenesis and vascular permeability [82,83,84,85], it is likely that elevation of arginase has an impact on these VEGF-mediated events. Conversely, studies in macrophages have shown that A1 expression is suppressed by expression or activation of VEGF receptor 1 (VEGFR1) [86]. These observations suggest a reciprocal relationship such that arginase may suppress VEGF function and vice versa.

Another experimental model with similarities to severe DR is the ischemia reperfusion (I/R) model [87]. The I/R model resembles the DR model in the formation of acellular capillaries as well as the biochemical profile. Studies using a mouse model of I/R found that A2 mRNA and protein levels were significantly increased within 3 h after I/R [88]. Conversely, levels of the cytosolic A1 isoform levels were significantly decreased after I/R and this was accompanied by decreased arginase activity [88,89]. The global deletion of A2 protected against the I/R-induced impairment of retinal function and neuronal degeneration and prevented the formation of acellular capillaries by reducing oxidative/nitrative stress and preventing glial activation [88]. On the other hand, the heterozygous deletion of A1 worsened the neurovascular degeneration after I/R, suggesting that A1 expression in some cells might play a protective role [89]. Further investigations suggested that A1 could be beneficial in preventing neurovascular injury after retinal I/R through affecting the macrophage inflammatory response and mediating repair as will be discussed in a later section.

### 3.3. Arginase in Premature Cellular Senescence

Cellular senescence is a state of cell cycle arrest in which the cells are viable but not dividing. Cellular senescence occurs physiologically during aging and is also a tumor suppressing mechanism [90]. However, cells can undergo senescence prematurely under stress conditions. For example, in conditions of limited metabolic supply, such as in ischemia, cellular senescence serves as a protective mechanism against increased energy demand.

Cellular senescence is associated with a complex network of reactions called senescence-associated secretory phenotype (SASP). SASP facilitates the autocrine and paracrine communication between cells to trigger inflammation and other detrimental effects [91]. An interesting study by Sapieha and colleagues has shown that retinal ischemia induces cell senescence in retinal ganglion cells in the OIR mouse model [92]. This was associated with induction of the SASP and spreading of cellular senescence to other cell types, including vascular cells undergoing pathological vitreoretinal neovascularization and retinal microglia [92]. They also observed cellular senescence in ganglion cells of STZ diabetic retinas, which suggested the involvement of premature senescence in DR. They confirmed this by measuring SASP factors in vitreous samples from patients and found significant increases in samples from patients with PDR as compared to control vitreous samples. Another study utilizing STZ diabetic rats found that senescence-associated factors are expressed mainly in diabetic retinal microvasculature and this was associated with increased oxidative and nitrative stress markers [93]. That study suggested oxidative/nitrative stress as a cause of premature senescence in diabetic retinas. Further investigation confirmed cellular senescence in human RECs subjected to HG which was associated with increased oxidative stress and mitochondrial dysfunction [94].

The diabetes-induced increase in arginase expression has also been linked to the induction of premature senescence. Investigations using in vivo and in vitro models of diabetes and A1 gene deletion or overexpression have strongly implicated arginase in diabetes-induced premature senescence [95]. The latter study showed that A1 gene deletion significantly protected against the increased activity of senescence associated β-galactosidase (SA β-gal) enzyme in isolated diabetic retinal vessels [95]. Conversely, the overexpression of A1 in REC increased cell stress along with premature senescence as shown by SA β-gal activity. In addition, the pharmacological inhibition of arginase activity significantly protected against the induced SA β-gal activity and decreased the mRNA expression of senescence-associated mediators, p16^INK4a^, p21, and p53 in diabetic retinas [95]. The investigators further demonstrated that diabetes or HG-induced activation of NOX2 and NOX2-derived ROS are upstream mediators of premature senescence in RECs by a mechanism involving activation of arginase [96]. That study, showed that the blockade of NOX2 or arginase reduced premature senescence in RECs, highlighting the role of arginase as a mediator of NOX2-induced premature senescence [96]. This process may contribute to the impaired vascular repair that is observed in diabetic conditions.

### 3.4. Arginase and Inflammation

Arginase not only contributes to oxidative stress, it also has been established to have a crucial role in inflammation. Almost two decades ago, the involvement of arginase in a murine model of endotoxin-induced uveitis was described [97]. A1 expression was found to be co-localized with iNOS and other L-arginine metabolizing enzymes in the infiltrated inflammatory cells in vitreous, ciliary body, iris, and inner retina. Later on, a study in a mouse model of uveitis found that glial and microglial A1 expression was significantly increased with LPS stimulation [98]. This study further showed that arginase deletion or inhibition significantly dampened leukostasis and suppressed inflammation, as demonstrated by reductions in tumor necrosis factor α (TNFα) and monocyte chemoattractant protein-1 (MCP-1) expression [98].

As noted above, several studies using diabetic murine models have confirmed the involvement of arginase in diabetes-induced retinal neuronal and vascular dysfunction and injury [37,75,96]. The A1 isoform is identified as a classic marker for anti-inflammatory M2-like macrophage [99]. On the other hand, the A2 isoform has been suggested to be present in pro-inflammatory M1-like macrophage [100]. In addition, A2 has been shown to induce inflammation in several different pathological models [101]. The specific role of macrophages in DR is not fully understood. However, an association of macrophages with DR pathology is well established [102]. Infiltrating macrophages have been found in significant amounts in vitreous samples collected from PDR patients [103]. Infiltrating macrophages were also detected in epiretinal membranes from PDR patients [104]. Arginase 1 expression can be detrimental or beneficial depending on the cell type. As explained above, increased A1 expression in endothelial cells is detrimental as it competes with endothelial NOS for L-arginine and decreases NO production, which is vital to maintain vascular function and to limit platelet aggregation and leukocyte attachment to the vessel wall. Upregulation of A1 also contributes to the increased oxidative stress in diabetic conditions. On the other hand, if arginase depletes L-arginine in macrophages, there will not be enough L-arginine available for iNOS. This will decrease iNOS function, limiting NO production and reducing iNOS transcription, which will in turn limit the inflammatory responses.

In the mouse model of retinal I/R, it has been reported that myeloid-specific deletion of A1 significantly worsens the I/R-induced neurodegeneration and retinal thinning [89]. Macrophages isolated from A1 knockout (A1KO) mice showed an increased inflammatory phenotype with LPS stimulation in vitro. On the other hand, treatment with PEGylated recombinant A1 (PEG-A1) protected against this inflammatory response to LPS stimulation. Moreover, intravitreal injection of A1 KO macrophages increased the neuronal injury after I/R. This study introduced A1 as a novel player in limiting neurovascular degeneration through stimulating macrophage-mediated repair [89].

### 3.5. Polyamine Metabolism in Retinopathy

Metabolism of polyamines (spermine, spermidine, and putrescine) involves combined actions of multiple enzymes [105]. Arginase plays a pivotal role in the polyamine biosynthesis pathway. Therefore, if arginase activity or expression is altered under diabetic conditions, polyamine metabolism is likely to be affected as well. Hence, polyamines can serve as downstream signaling mediators in the arginase pathway. Polyamine metabolism has been extensively studied in relation to oxidative damage and neurodegeneration [106,107,108].

However, little is known about the role of this pathway in DR. A study comparing polyamine levels in vitreous samples from patients with active PDR with samples from non-diabetic patients with proliferative vitreoretinopathy (PVR) showed that increases in spermine were positively correlated with increases in VEGF in patients with active PDR but not in those with PVR, suggesting the involvement of this polyamine in PDR [109]. Indeed, polyamine oxidation has been implicated in neuronal injury in both brain and retina [110,111]. The inducible SMOX enzyme in the polyamine catabolism pathway has also been linked to DR pathology [112]. In a study utilizing the STZ model of diabetes, the authors observed a significant increase in the expression of SMOX in the diabetic retina [113]. This study further investigated the impact of MDL 72527 (an inhibitor of SMOX) on diabetes-induced retinal neuronal damage and dysfunction. Treatment with MDL 72527 significantly improved retinal function in the diabetic mice and inhibited retinal thinning, RGC loss, and degeneration of inner retinal neurons, while reducing retinal levels of conjugated acrolein, an indicator of SMOX activity. These results support the involvement of SMOX in early signs of DR.

Because studies of polyamine metabolism in DR are limited, we can learn from studies conducted in different ischemic retinopathy models. Studies using the OIR model have shown that the expression of SMOX is significantly increased during the hyperoxia phase of OIR and that SMOX is a potential downstream signaling mediator of A2-induced neurovascular degeneration [114]. That study showed that hyperoxia-induced neuronal cell death was associated with increased hydrogen peroxide formation via activation of SMOX and that deletion of A2 prevented the SMOX activation, reduced the oxidative stress, and improved neuronal survival. The significant reduction in the level of spermine along with increased spermidine levels observed in the OIR retina was reversed in the A2 knock out retina, suggesting the role of A2 as an upstream mediator of this process [114]. In a follow-up study using the same OIR model, treatment with the polyamine oxidase inhibitor MDL 72527 significantly protected against the hyperoxia-induced vascular injury through a mechanism involving inhibition of microglia-mediated endothelial cell injury [115]. Blockade of SMOX using the same inhibitor also demonstrated neuroprotection in a model of retinal excitotoxicity, which mimics one of the mechanisms of neuronal damage in the diabetic retina [111].

Another aspect of the altered polyamine metabolism is the induction of vascular stiffness or fibrosis. In a high fat high sucrose (HFHS) mouse model of obesity and type 2 diabetes, endothelial-specific deletion of A1 protected against HFHS-induced aortic fibrosis and stiffness which was accompanied by reduced levels of plasma L-ornithine and less collagen formation [72]. In fibrotic vascular tissues collected from PDR patients, levels of the end product of unsaturated aldehyde acrolein-derived lipoxidation, FDP-Lys were significantly elevated implying a role for polyamine oxidation in this pathology [116]. The accumulation of FDP-Lys has been shown to contribute to abnormalities of Müller glia cells in an early DR model [117]. These results further support the potential involvement of the polyamine pathway in DR.

The main findings discussed in Section 3 about arginase and DR are summarized in Table 1 below.

## 4. Conclusions and Future Directions

In the current article, we review the existing literature about arginase and DR. We also have considered other experimental models of ischemic retinopathy like OIR and I/R due to the limited retinal injury seen in the current rodent models of DR. Studies to date have shown that arginase has an important role in the pathology associated with DR. Arginase is involved in both vascular and neuronal injury of DR through different mechanisms. Excessive A1 expression in RECs induces increases in oxidative stress and inflammation and promotes premature senescence. In contrast, A1 expression in immune cells can suppress inflammation. Moreover, administration of the PEG-A1 significantly protected against inflammation and neurovascular injury in the I/R model. More studies are still needed to determine the effect of PEG-A1 administration in diabetic models as well as in advanced stage neovascularization that occurs in PDR.

On the other hand, the deleterious role of A2 in retinal injury is well established. As discussed above, elevated levels of A2 are associated with retinal neurovascular degeneration in ischemic retinopathy models through mechanisms involving glial activation, increased oxidative stress, and alterations of polyamine pathways. In addition, the role of A2 in type 2 diabetes-induced retinal injury is beginning to emerge as noted above. A2 levels were significantly induced in retinas of a mouse model of obesity and type 2 diabetes that were accompanied by activation of inflammation pathways. Further studies are still needed to define the underlying mechanisms of A2-induced retinal the neurovascular retinal injury.

Mitochondrial dysfunction has been extensively studied in relation to the pathophysiology of DR. It has been reported that mitochondria can induce retinal damage in DR through increased oxidative stress, impaired energy production, and cell death. Those different mechanisms have been comprehensively reviewed previously [18,118,119]. One of the limitations of the current literature is that we do not have sufficient knowledge of whether arginase has an effect on mitochondrial function or not. As mentioned earlier, A2 is localized to the mitochondria. However, no studies have examined whether A2 is involved in mitochondrial function or dynamics in retinal diabetic models. In smooth muscle cells, overexpression of A2 increased mitochondrial ROS production and decreased the mitochondrial membrane potential, indicating mitochondrial dysfunction [120]. These effects were linked to vascular smooth muscles senescence and were independent of the enzymatic activity of arginase, adding another complex component to the role of arginase in disease [120]. Further studies of smooth muscle cells reported that A2 can decrease the mitochondrial membrane potential through regulating mitochondrial calcium uptake which is essential for the activation of p38 mitogen-activated protein kinase (MAPK) and interleukin 18 production [121]. Those effects were accompanied by increased levels of the polyamine spermine [121]. In endothelial cells subjected to hypoxia, which is a critical component of DR, the silencing of A2 prevented the hypoxia-induced mitochondrial superoxide production, monocyte adhesion, and the upregulation of intercellular adhesion molecule-1 (ICAM-1) levels [122].

Even though the studies outlined above reported a potential deleterious role of A2 in mitochondria, studies in other disease models have noted an opposite beneficial role of A2 in mitochondria. A study on a mouse model of asthma demonstrated that A2 knockout mice had a reduction in the mitochondria membrane potential. However, that effect was associated with worsened inflammation [123]. This latter study claimed a beneficial effect of A2 and arginine flux in preserving cell respiration capacity and bioenergetics in asthma [123]. These contradictory reports in different disease models underline the need for studies on the role of the arginase enzyme in mitochondrial function in specific models of DR.

Another future direction for DR research should be exploring the involvement of different pathways of energy metabolism in retina, by evaluating mitochondrial respiration and glycolysis as well as fatty acid oxidation in living tissue. This could be done by isolating ex-vivo retinal punches from different diabetic models and measuring the oxygen consumption rate and extracellular acidification rate using Seahorse XF analyzer instruments. Indeed, several studies using living tissues have examined cellular energetics in retinal hyperglycemia models [124,125,126]. However, further studies are still needed to confirm and elucidate the specific involvement of mitochondrial respiration versus glycolysis in these models and to determine how arginase can modulate these functions.

To summarize, even though the role of arginase is well established in the pathology that characterizes early DR, many aspects are waiting to be explored. Several questions still need answers, including:1-How is arginase regulated in DR?Further work is needed to address this question. Studies to date suggest that both isoforms are increased in retinal cells during DR but it is likely that these alterations occur in different cell types and that they are regulated by different mechanisms.2-What are the underlying mechanisms of arginase-induced retinal injury?Studies in different models suggest that multiple mechanisms contribute to the pathology, ranging from eNOS uncoupling in EC due to upregulation of A1 expression to suppression of microglial/macrophage-mediated reparative functions due to upregulation of A2 and suppression of A1 function. Alterations in polyamine metabolism could also be involved as a downstream mechanism of arginase-induced retinal injury. Studies to address these issues are in progress.3-What are the contradictory or complimentary roles of the two arginase isoforms?As has been outlined above, genetic studies have shown very different phenotypes with A1 vs. A2 deletion in that the former is lethal soon after birth whereas the latter produces only mild hypertension. Furthermore, studies in a variety of disease models have shown very different effects with A1 vs. A2 deletion in that A1 is prominently involved in promoting EC dysfunction whereas A2 is involved in microglia/macrophage-mediated inflammatory injury. Studies comparing the cell-specific effects of the two isoforms are needed to fully address this issue.4-Is arginase deleterious or beneficial in DR? Does arginase administration offer a therapeutic benefit for DR?

Studies to address these questions are in progress. The development of isoform-specific inhibitors and methods for cell-specific targeting of the different arginase pathways will greatly facilitate progress in this area. While efforts to develop isoform-specific inhibitors have been ongoing for a number of years, success has been limited up to now. This is largely due to the high homology between the two isoforms and to the fact that their active sites are identical. Given the differential actions of the two isoforms and their differential expression in different cell types, cell-specific delivery of general inhibitors may offer an alternative strategy for targeting the adverse effects of either isoform. Similarly, cell specific delivery of PEG-A1 to macrophage/microglial cells might offer an option for promoting its beneficial actions in limiting inflammation while avoiding its potentially detrimental effects on the vascular endothelium.

## Figures and Tables

**Figure 1 jcm-09-00425-f001:**
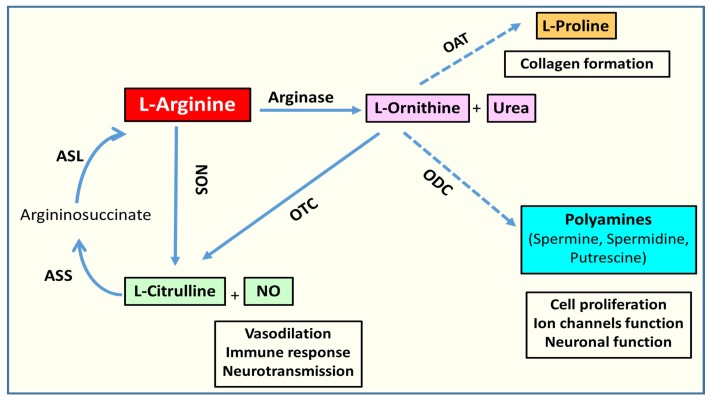
Schematic diagram of L-arginine metabolism. L-arginine is metabolized to L-ornithine and urea by arginase. It is also metabolized to L-citrulline and nitric oxide (NO) by the nitric oxide synthase (NOS). L-citrulline can be recycled back to L-arginine via the successive actions of argininosuccinate synthetase (ASS) and argininosuccinate lyase (ASL). L-ornithine can be converted to L-citrulline by the enzymatic action of ornithine transcarbamylase (OTC). L-ornithine can be used for polyamines production by ornithine decarboxylase (ODC). It can also be used for L-proline production by ornithine aminotransferase (OAT).

**Figure 3 jcm-09-00425-f003:**
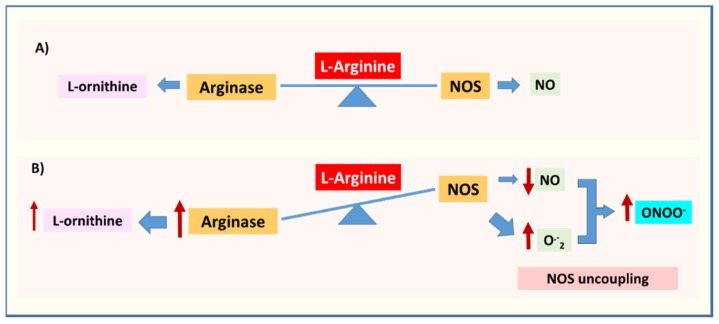
Arginase and Nitric oxide synthase (NOS) uncoupling. (**A**) Under physiological conditions both arginase and NOS compete for the same substrate L-arginine to produce L-ornithine and nitric oxide (NO), respectively. (**B**) Under pathological conditions increased arginase expression/activity can lead to NOS uncoupling and production of superoxide anion O^.-^_2_, which can react with NO to produce peroxynitrite (ONOO^−^). The figure is simplified to show the inverse relation between arginase and NOS.

**Table 1 jcm-09-00425-t001:** Table summarizing the main findings about arginase and diabetic retinopathy (DR) in animal models of ischemic retinopathies.

Animal Model	Main Findings	References
STZ Mice and Rats	Heterozygous deletion of A1 or arginase inhibition protected against the diabetes-induced retinal vascular dysfunction.	[75]
STZ Mice	A1 expression and activity are increased in 2-month diabetic retinas. The double knockout of A1 (one copy) and A2 (both copies) reduced superoxide production. Arginase inhibition enhanced NO formation and decreased leukocyte adhesion.	[37]
STZ Mice	Diabetes-induced activation of NOX2 and NOX2-derived ROS is linked to EC senescence through arginase activation. A1 gene deletion protected against the increased activity of senescence-associated β-galactosidase (SA β-gal) in isolated retinal vessels. The pharmacological inhibition of arginase activity decreased the expression of senescence-associated mediators.	[95,96]
STZ Mice	The expression of SMOX is increased in retinas of diabetic mice. The inhibition of SMOX activity improved retinal function and protected against the loss of inner retinal neurons.	[113]
HFHS Mice	A2 levels in the retina are increased after 16 weeks of Western diet. A2 deletion protected against the obesity-induced abnormalities in the ERG responses.	[76]
OIR Mice	A2 retinal levels are increased during the ischemic phase of OIR. A2 deletion ameliorated OIR-induced neurovascular alterations.	[79]
OIR Mice	The SMOX expression is increased during the hyperoxia phase of OIR, which was associated with neurovascular degeneration. A2 deletion prevented increases in SMOX expression/activity. The inhibition of SMOX activity protected against the hyperoxia-induced vascular injury through the inhibition of microglia-mediated endothelial cell injury.	[114,115]
I/R Mice	A2 expression was increased within 3 h after I/R. A1 expression and activity were significantly decreased after I/R. A2 deletion protected against I/R-induced neurovascular dysfunction. Conversely, the heterozygous deletion of A1 worsened the neurovascular injury.	[88,89]

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
