# Peer review of "Is the Arginase Pathway a Novel Therapeutic Avenue for Diabetic Retinopathy?"

_jcm, 2020, doi:10.3390/jcm9020425_

Round 1

Reviewer 1 Report

The authors have further improved their manuscript. They have addressed all of my previous issues and I have no further comments.

Reviewer 2 Report

thank you for addressing all comments. The review seems good now for publication. 

This manuscript is a resubmission of an earlier submission. The following is a list of the peer review reports and author responses from that submission.

Round 1

Reviewer 1 Report

This is a review paper by Shosha et al. looking into the literature on arginase-induced retinal neurovascular dysfunction in models of DR. This paper extensively discusses arginase and arginase pathway associated with Diabetic retinopathy looking at in vivo and in vitro studies. Major limitation would be that as mentioned in the study, there is not a good model of diabetic retinopathy in vivo. Most commonly used models for experiments are STZ induced hyperglycemia models which do not express any retinopathy or OIR models which do express neovascularization and nonperfusion however is not associated with hyperglycemia. Therefor none of the models will be able to resemble the true pathophysiology of diabetic retinopathy. This paper discusses about expression of arginase related polyamine pathway expression with active PDR, It would be interesting to see if there are any human studies with the retina or vitreous or metabolomics studies looking at the association of arginase pathway to support the association of this pathway with diabetic retinopathy.

Secondly, vascular and neuronal complications all share common root of metabolic dysfunctions leading to oxidative stress, however therapeutic approaches to alleviate stress on each of these pathways have generally been successful in laboratory experiments but only moderately effective in human clinical trials. In addition multiple antioxidant therapy regimes have proven ineffective, and in some cases diabetic patients on therapy were worse off than placebo. Therefore, it would be interesting to speculate more in  detail why the authors think arginase-pathway induced oxidative stress could be a therapeutic target for diabetic retinopathy. Moreover, given the detrimental and beneficial role of A1 isoform on diabetic retinopathy and inflammation, this would work against being a therapeutic target.

Thirdly, clinically diabetic retinopathy patients suffer from vision loss due to either diabetic macular edema or vitreous hemorrhage and tractional retinal detachment from advanced proliferative stage of diabetic retinopathy.  Both are strongly mediated by VEGF(vascular endothelial growth factor) upregulation causing ischemia and neovascularization. Although this paper focus on arginase pathway, would be interesting to comment if there is any relationship between the VEGF pathway and arginase pathway.

Reference 88:PVR and PDR has different pathophysiology and PVR is not associated with diabetes but associated with previous history of retinal detachment. According to this study, spermine content was up to 15 times higher only in vitreous from patients affected by PDR which also correlated with the VEGF level. Therefore spermine seems to the pivotal among the 3 different polyamines

Reviewer 2 Report

The review by Shosha et al. is a well written text providing an overview of the arginase pathway and its role in the neurovascular dysfunction underlying Diabetic Retinopathy (DR). Furthermore, they discuss the therapeutic potential arginase could play in the amelioration of DR.

The authors begin by providing the premise for the review by briefly introducing DR and the current therapeutic treatments for DR which are usually effective in limiting the late stage vascular pathology and thus the need for greater understanding of molecular mechanisms involved in the development of the diseases. They highlight the evidence showing that L-arginine metabolism needs to be tightly controlled as the arginase enzyme can affect oxidative stress, inflammation and cell death in DR. The authors subsequently discuss briefly arginase in diabetic models as well as other ischemic retinopathies. Furthermore, the authors highlight the role arginase plays in premature cellular senescence, inflammation and polyamine metabolism which all have a role in DR. As current available DR rodent models show limited retinal injury, they also discuss these concepts in relation to ischemic retinopathy models such as OIR and I/R. Additionally, they discuss the potential interesting role arginase could play in mitochondrial dysfunction and highlight this as an interesting avenue to explore in specific models of DR. Finally, they propose a number of open questions awaiting further investigation.

Although the focus of the manuscript is clearly on arginase and its future therapeutic potential a more balanced overview of other novel therapeutics for DR would greatly improve the manuscript. In addition, more information on the use of arginase inhibitors in diabetic patients already should be added. The authors should address other minor issues.

Major comments 

Is there any global statistics that could be added to the initial opening of the introduction? Not just on Diabetes itself but the prevalence of DR. What is the likelihood you get DR if you have Diabetes?

The Authors briefly discuss both laser photocoagulation and intravitreal injection of VEGF inhibitors. One or two sentences to further clarify the limitations of these current therapeutic strategies would be beneficial.

There are many other potential novel treatment avenues being explored for DR. Although the focus of this manuscript is on arginase, to provide a more balanced assessment it would be nice to add a separate small section or a paragraph in the discussion on these. For instance, the section or paragraph could include information on the following: the use of angiogenesis inhibitors; modulation of neurotrophic factors; Mesenchymal stem cells; Endothelial progenitor cells; gene therapeutic approaches: PEDF, VEGF, sFlt-1, Flt23k.

Although the role of arginase has been well established in diabetes and reviewed previously. Some more background would help the reader, with a particular focus on its use in patients to supplement the information given in animal models. This does not have to be extensive, but the authors could consider adding a small section or paragraph. As an example: Kovamees et al 2016 JCEM and similar work.

Section 3 would be helped with a table summarizing some of the findings from the animal models.

Some further speculation of potential redundancy/compensation between isoforms in should be added.

The authors end the discussion with a number of questions that need to be answered. It may be nice to have the authors own insight on how these might be addressed. E.g. Development of specific arginase inhibitors for the elucidation of the Isoforms contradictory or complimentary roles.

Minor comments

Check for needed hyphenation. E.g. anti-VEGF. Add “therapy” to Keywords. Eye surgery may also be needed in extreme cases of DR were laser photocoagulation is not possible. This should be mentioned in the introduction. Vitrectomy. Additionally, there is focal and panretinal photocoagulation. E.g. Line 45. Perhaps the authors could add some speculation on early diagnostics for DR? E.g. From the use of molecular markers or better imaging for improved screening. Line 54 “far, the treatments have been only partially effective and/or accompanied by adverse side effects [14-“ Remove “and/or” write out sentence: effective or accompanied by adverse effects or both. Line 80 , “ornithine decarboxylase (ODC) pathway for production of different polyamines through sequential” Add “the” in front of production. Line 124, “Thus, increased levels of arginase expression could be a tool for limiting inflammation in retina and” Add “the” before retina. Figure 1 is mentioned in Section 2.1 but is located in section 2.2. Figure 2 is mentioned in Section 2.2 but is located in section 2.3. Figure 2 Line numbering within figure. Line 132, The authors use the number 3 for number of isoforms then in Line 133 you use the word three. Change number 3 to the word three. Figure 3, you have an A and B part you do not mention these separate parts in the text or in the accompanying figure legend. Line 295, “depletes L-arginine in macrophages, there will be not be enough L-arginine available for iNOS. This” Fix “be not be enough”.